# Rational Design of Mono- and Bi-Nuclear Cyclometalated Ir(III) Complexes Containing Di-Pyridylamine Motifs: Synthesis, Structure, and Luminescent Properties

**DOI:** 10.3390/molecules27186003

**Published:** 2022-09-15

**Authors:** Hugo Sesolis, Geoffrey Gontard, Marie Noelle Rager, Elisa Bandini, Alejandra Saavedra Moncada, Andrea Barbieri, Hani Amouri

**Affiliations:** 1Institut Parisien de Chimie Moléculaire (IPCM) UMR CNRS 8232, Sorbonne Université-Campus Pierre et Marie Curie, 4 place Jussieu, CEDEX 05, 75252 Paris, France; 2Chimie ParisTech, NMR Facility, PSL University, 75005 Paris, France; 3Istituto per la Sintesi Organica e la Fotoreattività (ISOF), Consiglio Nazionale delle Ricerche (CNR) Via Gobetti 101, 40129 Bologna, Italy

**Keywords:** phosphorescent iridium complexes, blue emitters, X-ray structural determination

## Abstract

Heteroleptic cyclometalated iridium (III) complexes (**1**–**3**) containing di-pyridylamine motifs were prepared in a stepwise fashion. The presence of the di-pyridylamine ligands tunes their electronic and optical properties, generating blue phosphorescent emitters at room temperature. Herein we describe the synthesis of the mononuclear iridium complexes [Ir(ppy)_2_(DPA)][OTf] (**1**), (ppy = phenylpyridine; DPA = Dipyridylamine) and [Ir(ppy)_2_(DPA-PhI)][OTf] (**2**), (DPA-PhI = Dipyridylamino-phenyliodide). Moreover, the dinuclear iridium complex [Ir(ppy)_2_(**L**)Ir(ppy)_2_][OTf]_2_ (**3**) containing a rigid angular ligand “**L** = 3,5-bis[4-(2,2′-dipyridylamino)phenylacetylenyl]toluene” and displaying two di-pyridylamino groups was also prepared. For comparison purposes, the related dinuclear rhodium complex [Rh (ppy)_2_(**L**)Rh(ppy)_2_][OTf]_2_ (**4**) was also synthesized. The x-ray molecular structure of complex **2** was reported and confirmed the formation of the target molecule. The rhodium complex **4** was found to be emissive only at low temperature; in contrast, all iridium complexes **1**–**3** were found to be phosphorescent in solution at 77 K and room temperature, displaying blue emissions in the range of 478–481 nm.

## 1. Introduction

Cyclometalated iridium (III) complexes are an important class of molecules because they display remarkable photophysical properties [1] for a wide range of applications [2,3,4,5,6,7]. Moreover, they show high stability, which makes them adequate emitters for organic light-emitting devices (OLEDs) [8,9,10]. Generally, they display octahedral geometry with six coordination bonds and can be obtained as neutral, homoleptic species, such as [Ir(C^N)_3_], with three cyclometalated ligands bound to the metal center [11,12]. For instance, [Ir(ppy)_3_] is a well-representative molecule for such compounds and behaves as a green emitter for organic light-emitting devices with high efficiency as reported by Thompson and coworkers [13]. The introduction of a bipyridine (bpy)-type ligand (N^N) to the metal center generates cationic heteroleptic complexes of the type [Ir(C^N)_2_(N^N)]^+^ [14,15,16]. Such compounds have been utilized as light-emitting-electrochemical cells (LECs) [17,18]. Modifying the nature of the ancillary ligands affects the optical properties of the compounds and makes them emissive throughout the whole range of the electromagnetic visible and NIR spectrum [19,20,21,22]. Replacement of the bipyridine by the dipyridylamine (DPA) ligand also provides cationic complexes of the type [Ir(C^N)_2_(DPA)]^+^; however, it affects the electronic properties of the molecule relative to the [Ir(C^N)_2_(N^N)]^+^ complexes (C^N = ppy; N^N = bpy) [23]. Unlike bpy, the two pyridyl units in DPA are not conjugated, and this contributes to push the LUMO level to higher energy without changing the HOMO (Figure 1). As a consequence, the HOMO–LUMO gap becomes larger, and eventually, such compounds become more suited to emit at higher energy, when compared to their analogous compounds [Ir(C^N)_2_(N^N)^+^ with bipyridine ligands (C^N = F2ppy, ppy; N^N = bpy) [24,25]. Hence, using the DPA ligand might be an interesting approach to prepare complexes that emit in the blue region. Pursuing our research activity in this area of luminescent metal complexes, we prepared several cyclometalated rhodium and iridium complexes (**1**–**4**) (Figure 2) containing dipyridylamine motifs in a stepwise fashion. The purpose of this work is to investigate the effect of the functionalized dipyridyl amine on the optical properties of these complexes. Furthermore, the binuclear rhodium and iridium complexes containing rigid angular ligands displaying two dipyridylamine motifs are also reported, including their luminescent properties when compared to the related mononuclear species.

## 2. Results and Discussion

### 2.1. Synthesis and Characterization

Complexes **1**–**4** were obtained in two steps, following a synthetic procedure developed by our group. The first step consisted of the preparation of the solvated metal complex in situ, by treatment of the chloride dimer [M(ppy)_2_(-Cl)]_2_ (M = Rh, Ir), with two equivalents of AgCF_3_SO_3_ in acetone at room temperature for 2 h. Then, the mixture was filtered to remove the solid AgCl and subsequently treated with the desired ligand in dichloroethane under reflux for 12 h, followed by the reaction work up afforded the target compounds **1**–**4** in good yields, ranging from 71 to 95% (Figure 2).

The ^1^H-NMR spectra of complexes **1**–**4** recorded in CD_2_Cl_2_ and CD_3_NO_2_ confirm the formation of the target complexes. For instance, complex **1** displayed a symmetric pattern for the aromatic protons of the two “ppy” and the dipyridyl amino groups, due to the presence of *C*2-symmetry in the molecule. A total of 10 multiplets were visible in the range of δ 6.1–8.3 ppm; moreover, the amine-H appeared downfield at δ 9.92 ppm. On the other hand, compound **2**, which also contains a *C*2-symmetry, displayed 12 multiplets for the aromatic protons between δ 6.1 ppm and δ 8.0 ppm, highlighting a symmetric motif as well. As for the binuclear complexes, the ^1^H-NMR of **3** and **4** displayed similar spectra. Again, a well symmetric pattern was visible for the aromatic protons, where 13 multiplets for **3** (11 multiplets for **4**) appeared in the range of δ 6.2 ppm to δ 8.2 ppm, in addition the methyl protons that appeared as a singlet at δ 2.41 ppm, which was downfield relative to the free ligand L. The ^13^C-NMR spectra of complexes **3** and **4** showed 24 aromatic carbons in the range δ 116.8–167.4 ppm, confirming the symmetric pattern observed in the solution for these molecules. Moreover, the integrity of the iridium and rhodium dimers **3**–**4** in the solution was ascertained by electrospray spectrometry, in which [{Ir(ppy)_2_}_2_(L)]^2+^ fragment was identified at *m*/*z* = 816.2208, and for [{Rh(ppy)_2_}_2_(L)]^2+^, at *m*/*z* = 726.1628, respectively (Appendix A). A complete spectroscopic characterization (^1^H, ^13^C, IR, MS) and elemental analyses are given in the experimental section and SI (Appendix A). Moreover, the structure of **2** was confirmed by a single-crystal X-ray diffraction study. 

### 2.2. X-ray Molecular Structure of [Ir(ppy)_2_(DPA-PhI)][OTf] (**2**)

Convenient crystals of **2** were obtained from CH_2_Cl_2_/Et_2_O via a slow evaporation process of diethyl ether into a CH_2_Cl_2_ solution of the complex. The structure confirms the formation of the target molecule **2** (Figure 1a). 

The structure shows that the iridium center is coordinated to two “ppy” ligands and one dipyridylamino moiety, generating a distorted octahedral geometry around the metal center. The two nitrogen centers of the two ppy units were disposed in *trans*-configuration, with N-Ir bond distances of 2.047(3) Å and 2.062(3) Å. The two carbon centers were in *cis*-geometry and induced a *trans*-effect on the two nitrogen centers of the DPA-PhI ligand. As a consequence, the N-Ir bond distances were 2.168(3) Å and 2.172(3) Å. These distances are longer than those observed for the two ppy units. This data is in accord with those reported previously. The chelating angle of the dipyridylamino ligand (N^N) towards the Ir(III) center was 83.82(10)°, as expected for a six-membered metallacycle. This is larger than those of the ppy ligands (C^N), 80.27(12)° and 80.49(12)°. Unlike the (C^N) ligands, which are planar, the N^N chelate ring adopted a boat conformation with central amine nitrogen bonded to the -PhI group. The latter was disposed of in the apical position (Figure 1a). 

Examining the packing of the molecules in the crystal revealed the presence of two sets of π-π interactions among individual molecules (Figure 1b). The first interaction occured between two ppy units of the two adjacent molecules, with π-π contacts with C20···C24 at 3.342(7) Å, while the second π-π interaction with C8···C9 at 3.644(5) Å occured between two diphenylamino groups of two close individual molecules generating 1D supramolecular assembly.

### 2.3. Absorption Properties

The absorption spectra of the ligand the **L** and the complexes **1–4** in the CH_3_CN solution at room temperature are depicted in Figure 2, and related key data are reported in Table 1.

The ligand **L** exhibited a broad and relatively intense absorption band and two shoulders between 280 and 340 nm in the UV region. The first one was centered at about 336 nm, with ε = 68,800 M^−1^ cm^−1^ and one of the shoulders at 318 nm (ε = 65,200 M^−1^ cm^−1^). These two lowest-energy bands are attributed to the amine N-to ring CT transitions. The second shoulder, located at a higher energy at ~286 nm (ε = 45,400 M^−1^ cm^−1^), was assigned to the pyridine-based ligand-centered (LC) transition. These features were also observed in previous studies of 2,2′-dypyridylamine derivatives [26,27].

The Ir(III) complexes **1–3** presented several absorption bands in the UV region (Figure 2). The most intense absorption bands were found between 245 to 265 nm, in which the molar absorption coefficient of the binuclear complex **3** was twice as intense (ε = 104,000 M^−1^ cm^−1^) as the mononuclear complexes **1** and **2** (ε < 49,000 M^−1^ cm^−1^). The absorption behavior of [Ir(ppy)_2_(DPA)][OTf] (**1**), as well as its luminescence properties (*vide infra*), were almost identical to what was already reported for the analogous [Ir(ppy)_2_(DPA)][PF_6_], with a hexafluorophosphate counter ion [23]. Additionally, complex **3** showed another three intense and defined absorption bands at 295, 306, and 336 nm (ε > 69,000 M^−1^ cm^−1^), while the mononuclear complexes exhibited only two shoulders between 260 and 300 nm (ε < 42,000 M^−1^ cm^−1^). These absorption bands are associated with the spin-allowed π-π* ligand-centered (LC) transitions from dipyridylamine and phenylpyridine ligands [25,26,28,29,30]. For all the complexes, a weak and broad absorption band above 370 nm was observed, which is attributed to metal-to-ligand charge transfer (MLCT) transitions, along with ligand-centered (LC) contribution from the ligands. In addition to these transitions, a long tail extended to the visible region can be attributed to the strong spin-orbit coupling (SOC) applied by the Ir atom, which can cause the direct singlet–triplet transition [2,6,23,31,32,33]. The Rh(III) derivative **4** displayed a similar envelope of absorption bands, with the obvious absence of the low energy singlet–triplet direct absorption above 380 nm [34].

### 2.4. Luminescence Properties

The photoluminescence properties of the bridging ligand **L** and complexes **1–4** were investigated in de-aerated and air-equilibrated CH_3_CN solution at room temperature (298 K) and in CH_3_OH:C_2_H_5_OH (1:4) glassy solution at 77 K. The normalized emission spectra of the ligand and complexes at 298 K are shown in Figure 3, and the spectra at 77 K are given in Figure 4. The emission maxima (λ_max_), photoluminescence quantum yields (ϕ), and lifetimes (τ) are collected in Table 2.

The ligand **L** exhibits at room temperature a relatively broad and unstructured emission band, centerd at 395 nm (Figure 3). Notably, **L** presents a high quantum yield in de-aerated solution (ϕ = 73.7%), moderately quenched by oxygen in aerated solution (ϕ = 64.0%), and shortens lifetimes on the nanosecond time scale, even when the oxygen is removed (τ = 1.6 ns in de-aerated and 1.4 ns in aerated solution, respectively). This behavior, which can be attributed to the fluorescence of the 2,2′-pyridylamine units of the molecule, is in line with previous studies of parent compounds and their use as blue-emitting active material in OLED devices, as has been suggested [26,35,36,37,38,39,40]. The fluorescence spectrum of **L** in CH_3_OH: C_2_H_5_OH glassy solution at 77 K is slightly more structured and hypsochromically shifted about 2300 cm^−1^, with respect to that at room temperature, and presents a shorter lifetime, τ = 0.7 ns (Figure 4, Table 2). Interestingly, under the same conditions and under pulsed excitation light in time-gated detection mode, the phosphorescence emission of the ligand **L** has also been detected, displaying a structured emission profile with a maximum at about 494 nm, ca. 20,250 cm^−1^ (Figure 4).

The Ir(III) complexes **1–3** presented in the de-aerated CH_3_CN solution at room temperature a significant red-shift of the emission, in comparison to the free ligand (Figure 3), exhibiting broad, structured, and almost identical emission spectra, with λ_max_ at about 480 nm, a second band at ~520 nm, and a shoulder around 550 nm. The novel dinuclear Rh(III) complex [Rh (ppy)_2_(**L**)Rh(ppy)_2_][OTf]_2_ (**4**) displayed a non-emission behavior. We also note that the mononuclear complex containing dipyridylamine [Rh(ppy)_2_(DPA)][PF_6_] was also found to be non-emissive [34]. The vibronic structure observed indicates a predominantly ^3^LC π-π* character of the emissive excited state, rather than ^3^MLCT or ^3^LLCT characters. This is in line with the photoluminescence behavior of other cyclometalated Ir(III) complexes [29,30,41,42,43,44,45,46]. However, differences in the photoluminescence quantum yields and lifetimes can be seen between them in Table 2. In de-aerated CH_3_CN solutions, the mononuclear complex **1** exhibited a remarkably higher quantum yield and longer lifetime, ϕ = 11.0% and τ = 510 ns, than the complex **2** (ϕ = 0.3%, τ = 20 ns). This observation indicates that adding a phenyl unit with an iodine atom to the ligand 2,2′-diphenylamine has a detrimental effect on the quantum yield, and the excited-state lifetime concomitantly decreases. The same reasoning can be applied to the binuclear complex **3**, where its emission properties (ϕ = 0.6%, τ = 110 ns) were affected by the increment of the conjugation of the ligand and the role that plays the formed bridge. Similarly, Williams and co-workers had observed that the complex [Ir(L^4^)_2_]^3+^ [43], and the binuclear complex [Ir(tpy)(μ-tpy-ϕ-ϕ-mtbpy)Ir-(dpyx)]^4+^ [47], presented short lifetimes and low quantum yields. This was attributed to an increment of the conjugation by introducing a bis-mesityl group into the terpyridine ligand for the former, and to an extended bridging ligand for the latter complex. These results have already been seen in other families of binuclear Ir(III) complexes, where lower quantum yields than their related mononuclear derivatives were shown, due to the nature of the bridging ligands and the length of the spacer [1,48,49,50,51,52,53]. In our case, the lower quantum yield and shorter emission lifetimes suggest an enhancement of the non-radiative decay pathways.

In air-equilibrated CH_3_CN solutions, the three Ir(III) complexes **1**–**3** were less emissive, the quantum yield being about two times lower than in de-aerated solutions. At the same time, the emission lifetimes were shortened; complex **1** presents a 10-fold decrease, while complex **3** displays a smaller quenching by oxygen, ca. a 2-fold decrease. The phosphorescence emission of complex **2**, bearing the phenyl-iodine group in the dipyridylamine ligand, was only slightly quenched by oxygen, suggesting that the excited triplet-state de-activation back to the ground singlet-state was dominated by the non-radiative intersystem crossing processes promoted by the strong spin-orbit coupling induced by the iodine atom (ζ = 5069 cm^−1^), rather than the triplet–triplet energy transfer to the molecular oxygen [54].

All investigated complexes **1**–**4** and the bridging ligand **L** were luminescent in CH_3_OH:C_2_H_5_OH glassy solution at 77 K. The three Ir(III) complexes **1**–**3** showed emission spectra that were much more structured, with respect to the emission in the CH_3_CN solution at 298 K. As observed in Figure 4, the emission bands of the Ir(III) complexes were similar to the phosphorescence emission of the ligand **L**, suggesting that the emissive excited states of the complexes originated from the dipyridylamine unit of the ligand. Nonetheless, the emission spectra of the binuclear complex **3** were more similar to that of the ligand **L** and exhibited a maximum emission at 488 nm, with small rigidochromic blue-shift (6 nm) with respect to that of the ligand **L**, and was red-shifted (~10 nm) compared to its emission in CH_3_CN solution at 298 K. On the other hand, the two mononuclear complexes, **1** and **2**, exhibited an emission maximum at about 470 nm, and a small blue-shift was observed, around 8 nm, with respect to their emission in CH_3_CN at 298 K. These observations can confirm that the ^3^LC π-π* character is predominant in the emissive excited state [11,29,31,41,55]. The Rh(III) derivative **4** shows a structured emission, blue-shifted with respect to that of the corresponding Ir(III) derivative **3** by ca. 1060 cm^−1^. In this case the emission can also be attributed to ^3^LC excited states, as for the Ir(III) analogue **3** [56,57].

The trend of emission lifetimes of the Ir(III) complexes **1**–**3** differed substantially from 298 K to 77 K (Table 2). In a glassy solution at 77 K, longer mono-exponential lifetimes were observed. This feature is commonly observed in Ir(III) ortho-metalated complexes, as the rigid matrix and the low temperature usually hinder the non-radiative mechanisms, thus increasing the luminescence intensity and the relevant excited-state lifetime [1]. While the mononuclear complexes exhibited lifetimes in the range of the microseconds, τ = 4.8 μs for **1** and τ = 5.2 μs for **2**, which are in line with previous findings of other mononuclear Ir(III) complexes [29,31,41,55,58,59,60,61], the binuclear complex **3** showed significantly longer emission lifetimes, in the range of the milliseconds: τ = 5.5 ms. As mentioned before, the extended conjugation and the presence of substituents on the ligands play a significant role in the photophysics of the complexes. It should be noted that relatively long lifetimes in the microseconds scale have been found in other families of binuclear Ir(III) complexes with different bridging ligands [31,47,48,49,62,63,64,65], but excited state lifetimes in the millisecond scale are relatively rare for Ir(III) complexes [1].

## 3. Conclusions

In this paper, we described the synthesis of some mononuclear cyclometalated iridium complexes (**1**–**2**) containing dipyridylamine ligands. The x-ray molecular structure of complex **2** is reported and confirms the formation of the target molecule. Moreover, we successfully extended our synthetic methodology and prepared two rare binuclear cyclometalated iridium (**3**) and rhodium (**4**) complexes featuring two dipyridylamine motifs. These binuclear complexes represent the first examples of these types of compounds. The Ir(III) complexes **1**–**3** displayed a blue phosphorescence at about 480 nm, originating from ^3^LC excited states mainly located on the bridging ligand bearing the dipyridylamine moiety. The Rh(III) complex **4** was only luminescent in glassy solution at 77 K, with the emission originating from ^3^LC excited states as well. The dinuclear complex **3** presented an unusually long excited-state lifetime, in the millisecond range, for this kind of complex. Notably, in the bridging ligand **L,** the fluorescence and phosphorescence emissions were both observed. The dipyridylamine ligand evidenced a strong influence on the photophysical properties of the iridium complexes.

## 4. Materials and Methods

All solvents used were of reagent grade or better. Deuterated solvents and commercially available reagents were used as received, unless otherwise specified. The ^1^H-NMR spectra were recorded on Bruker Avance-400 and Avance-Neo 500 spectrometers. Chemical shifts were reported in the ppm downfield from tetramethylsilane and refer to the residual hydrogen signal of deuterated solvents (CHD_2_NO_2_ at 4.33 ppm, CHDCl_2_ at 5.32 ppm) and the residual solvent carbon signal (CD_3_NO_2_ at 61.4 ppm, CD_2_Cl_2_ at 53.5 ppm) for ^13^C NMR. IR spectra were recorded on a Bruker Tensor 27 equipped with a Harrick ATR. Ligand 3,5-Bis[4-(2,2′-dipyridylamino)phenylethynyl] toluene (**L**) was prepared, as reported previously by our group [66].

**[Ir(ppy)_2_(DPA)][OTf] (1).** [Ir(ppy)_2_(μ-Cl)]_2_ (75 mg, 0.07 mmol) and AgOTf (39 mg, 0.15 mmol.) were introduced into a Schlenk tube containing acetone (20 mL), and the mixture was refluxed for 2 h. Then the suspension was filtered through celite to remove the AgCl precipitate and the filtrate was collected and dried under vacuum. To this was added, via a cannula, a solution of dipyridylamine (DPA) (26 mg, 0.15 mmol) in dichloroethane (10 mL). The reaction mixture was heated to reflux with stirring for 12 h and then cooled to room temperature. The solution was concentrated under vacuum, and the subsequent addition of diethyl ether (50 mL) created a yellow precipitate. The resulting precipitate was collected by filtration, washed with ether (2 × 10 mL), and dried under a vacuum to make complex **1** a yellow powder (102 mg, 83%). The ^1^H NMR (500 MHz, CD_2_Cl_2_) *δ* (ppm) found: 9.92 (s, 1H, NH), 8.21 (ddd, *J* = 5.9; 1.5; 0.7 Hz, 2H, H2), 7.95 (br d, *J* = 8.2 Hz, 2H, H5), 7.84 (ddd, *J* = 8.2; 7.5; 1.5 Hz, 2H, H4), 7.70–7.65 (m, 4H, H8 and H15), 7.60–7.57 (m, 4H, H13 and H16), 7.15 (ddd, *J* = 7.5; 5.9; 1.5 Hz, 2H, H3), 6.98 (ddd, *J* = 7.8; 7.3; 1.2 Hz, 2H, H9), 6.84 (td, *J* = 7.5; 1.4 Hz, 2H, H10), 6.62 (ddd, *J* = 7.2; 5.9; 1.4 Hz, 2H, H14), 6.17 (br. d, *J* = 7.5 Hz, 2H, H11). The ^13^C NMR (125 MHz, CD_2_Cl_2_) *δ* (ppm): 167.7 (C6), 151.7 (C17), 150.0 (C12), 149.6 (C13), 149.5 (C2), 144.0 (C7), 139.3 (C15), 138.2 (C4), 131.8 (C11), 130.4 (C10), 124.8 (C8), 123.0 (C3), 122.4 (C9), 119.8 (C5), 119.0 (C14), 116.3 (C16). The results were calculated for C_33_H_25_F_3_N_5_O_3_IrS.H_2_O: C 47.25; H 3.24; N 8.35%. We found: C 47.35; H 3.22; N 8.22%. IR, *v*(CF_3_SO_3_^−^) 1030 cm^−1^; 1251 cm^−1^.

**[Ir(ppy)_2_(DPA-PhI)][OTf] (2).** This compound was prepared in a similar way to that described for [Ir(ppy)_2_(DPA)][OTf], but using [Ir(ppy)_2_(μ-Cl)]_2_ (100 mg, 0.092 mmol), AgOTf (52 mg, 0.20 mmol.) and ligand 4-(2,2′-Dipyridylamino)-phenyl iodide (DPA-PhI) (75 mg, 0.20 mmol.). The target compound was isolated as a yellow powder (164 mg, 81%). The ^1^H NMR (500 MHz, CD_2_Cl_2_) *δ* (ppm) found: 7.97 (ddd, *J* = 8.2; 1.3; 0.8 Hz, 2H, H5), 7.94–7.90 (m, 4H, H13 and H15), 7.88 (ddd, *J* = 5.8; 1.5; 0.8 Hz, 2H, H2), 7.82 (ddd, *J* = 8.2; 7.5; 1.5 Hz, 2H, H4), 7.70 (dd, *J* = 7.8; 1.5 Hz, 2H, H8), 7.67 (d, *J* = 9.0 Hz, 2H, H21), 7.58 (dd, *J* = 8.9; 1.3 Hz, 2H, H16), 7.07 (ddd, *J* = 7.2; 5.9; 1.3 Hz, 2H, H14), 7.02 (ddd, *J* = 7.8; 7.3; 1.2 Hz, 2H, H9), 6.90–6.86 (m, 4H, H3 and H10), 6.65 (d, *J* = 9.0 Hz, 2H, H20), 6.21 (ddd, *J* = 7.6; 1.2; 0.4 Hz, 2H, H11). The ^13^C NMR (125 MHz, CD_2_Cl_2_) *δ* (ppm) found: 167.6 (C6), 153.5 (C17), 151.5 (C13), 149.7 (C2), 148.1 (C12), 143.9 (C7), 143.2 (C19), 140.7 (C15), 139.4 (C21), 138.5 (C4), 131.7 (C11), 130.5 (C10), 125.0 (C8), 123.7 (C14), 123.2 (C16), 122.9 (C9), 122.8 (C3), 121.9 (C20), 120.0 (C5), 88.8 (C22). The results were calculated for C_39_H_28_F_3_IN_5_O_3_IrS.H_2_O: C 45.00; H 2.91; N 6.73%. We found: C 44.68; H 2.88; N 6.66%. IR(ATR), *v*(CF_3_SO_3_^−^) 1030 cm^−1^; 1251 cm^−1^.

**[{Ir(ppy)_2_}_2_(L)][OTf]_2_ (3).** Complex **3** was obtained following a similar procedure as described above, but starting with the following amounts [Ir(ppy)_2_(μ-Cl)]_2_ (65.0 mg, 0.06 mmol), AgOTf (35 mg, 0.13 mmol.), and 3,5-Bis[4-(2,2′-dipyridylamino)phenylethynyl]toluene (**L**) (40 mg, 0.06 mmol). Compound **3** was obtained as a yellow powder (114 mg, 95%). The ^1^H NMR (400 MHz, CD_3_NO_2_) *δ* (ppm) found: 8.12–8.06 (m, 12H, H5, H13 and H15), 8.02 (d, *J* = 5.9 Hz, 4H, H2), 7.95–7.92 (m, 4H, H16), 7.88 (td, *J* = 8.0; 1.5 Hz, 4H, H4), 7.79 (dd, *J* = 7.8; 1.0 Hz, 4H, H8), 7.54 (br.s, 1H, H26), 7.49 (d, *J* = 9.0 Hz, 4H, H21), 7.43 (br. S, 2H, H27), 7.27–7.22 (m, 4H, H14), 7.00 (td, *J* = 7.8; 1.1 Hz, 4H, H9), 6.93 (d, *J* = 9.0 Hz, 4H, H20), 6.90–6.84 (m, 8H, H3 and H10), 6.32 (dd, *J* = 7.8; 1.1 Hz, 4H, H11), 2.41 (s, 3H, H29). The ^13^C NMR (100 MHz, CD_3_NO_2_)) *δ* (ppm) found: 167.3 (C6), 153.5 (C17), 151.5 (C13), 150.3 (C2), 148.4 (C12), 144.6 (C7), 144.5 (C19), 140.9 (C15), 139.5 (C28), 138.4 (C4), 133.0 (C21), 131.9 (C27), 131.8 (C11), 131.0 (C26), 130.0 (C10), 124.8 (C16), 124.7 (C8), 124.4 (C14), 123.5 (C25), 122.8 (C3), 122.7 (C9), 119.8 (C5), 117.5 (C22), 116.9 (C20), 88.9 (C23), 88.4 (C24), 19.8 (C29). IR(ATR), *v*(CF_3_SO_3_^−^) 1032 cm^−1^; 1253 cm^−1^. The ES-HRMS (*m*/*z*) was: [{Ir(ppy)_2_}_2_(L)]^2+^:816.2203; the result found was: 816.2208.

The results were Calculated for C_89_H_62_F_6_N_10_O_6_IrS_2_.C_2_H_4_Cl_2_: C 53.87; H3.28; N 6.90%. We found: C 53.90; H 3.14; N 7.13. IR,*v*(CF_3_SO_3_^−^) 1030 cm^−1^; 1251 cm^−1^. 

**[{Rh(ppy)_2_}_2_(L)][OTf]_2_ (4).** This compound was prepared in a similar procedure to that described for [Ir(ppy)_2_(DPA)][OTf], but using the following materials: [Rh(ppy)_2_(μ-Cl)]_2_ (53 mg, 0.06 mmol), AgOTf (34 mg, 0.13 mmol.), and 3,5-Bis[4-(2,2′-dipyridylamino)phenylethynyl]toluene (**L**) (40 mg, 0.06 mmol). Compound **4** was obtained as an off-white powder (78 mg, 71%). The ^1^H NMR (400 MHz, CD_3_NO_2_) *δ* (ppm) found: 8.14–8.08 (m, 8H, H5 and H13), 8.08–8.02 (m, 8H, H2 and H15), 7.96 (ddd, *J* = 8.1; 7.5; 1.5 Hz, 4H, H4), 7.86–7.80 (m, 8H, H8 and H16), 7.54 (br.s, 1H, H26), 7.50 (d, *J* = 9.0 Hz, 4H, H21), 7.43 (br. s, 2H, H27), 7.25 (ddd, *J* = 7.2; 5.7; 1.2 Hz, 4H, H14), 7.08 (td, *J* = 7.6; 1.2 Hz, 4H, H9), 6.98–6.90 (m, 12H, H3, H10 and H20), 6.37 (d, *J* = 7.6 Hz, 4H, H11), 2.41 (s, 3H, H29). The ^13^C NMR (100 MHz, CD_3_NO_2_)) *δ* (ppm) found: 166.3 (d, *J_CRh_* = 33.0 Hz, C12), 164.6 (C6), 154.0 (C17), 151.2 (C13), 150.3 (C2), 144.5 (C19), 144.4 (C7), 141.0 (C15), 139.5 (C28), 138.6 (C4), 133.1 (C21), 132.9 (C11), 131.9 (C27), 131.0 (C26), 129.9 (C10), 124.6 (C8), 124.1 (C16), 123.5 (C9, C14 and C25), 122.9 (C3), 120.0 (C5), 117.75 (C20), 117.66 (C22), 88.9 (C23), 88.4 (C24), 19.8 (C29). The results were Calculated for C_89_H_62_F_6_N_10_O_6_Rh_2_S_2_.3/2C_2_H_4_Cl_2_:C 58.16; H 3.61; N 7.37%. We found: C 58.18; H 3.32; N 7.45%. IR,*v*(CF_3_SO_3_^−^) 1030 cm^−1^; 1251 cm^−1^. The ES-HRMS (*m*/*z*) was: [{Rh(ppy)_2_}_2_(L)]^2+^:726.1629; we found: 726.1628.

**X-Ray crystal structure determination.** A single crystal was selected, mounted, and transferred into a cold nitrogen gas stream. Intensity data was collected with a Bruker Kappa-APEX2 system, using fine-focus sealed tube Mo-Kα radiation. Unit-cell parameters determination, data collection strategy, integration, and absorption correction were carried out with the Bruker APEX2 suite of programs. The structure was solved with SIR97 and refined anisotropically by full-matrix least-squares methods with SHELXL, using WinGX. The structure was deposited at the Cambridge Crystallographic Data Centre with number CCDC 2171075 and can be obtained free of charge via www.ccdc.cam.ac.uk (accessed on 8 September 2022).

**Crystal data for 2:** C_39_H_32_F_3_IIrN_5_O_5_S, triclinic P -1, a = 9.3487(3) Å, b = 14.9683(5) Å, c = 15.8197(5) Å, α = 95.105(2)°, β = 106.778(1)°, γ = 105.650(1)°, V = 2007.20(11) Å^3^, Z = 2, green bar 0.5 × 0.1 × 0.05 mm^3^, μ = 4.209 mm^−1^, min/max transmission = 0.42/0.49, T= 200(1) K, λ = 0.71073 Å, θ range = 2.16° to 30.54°, 59351 reflections measured, 12284 independent, R_int_ = 0.0193, completeness = 0.999, 514 parameters, 0 restraints, final R indices R1 [I > 2σ(I)] = 0.0310 and wR2 (all data) = 0.0885, GOF on F^2^ = 1.111, and largest difference peak/hole = 2.70/−1.13 e·Å^−3^.

**Photophysical measurements**. All solvents used for photophysical studies were of spectroscopic grade and were used without further purification. Square optical Suprasil Quartz (QS) cuvettes of 1 cm path length were used for the absorption and emission measurements at room temperature. Luminescence measurements of CH_3_OH:C_2_H_5_OH (1:4) frozen glassy solutions at 77 K were performed in quartz capillary tubes immersed in liquid nitrogen, hosted within a homemade quartz cold finger Dewar.

The absorption spectra of dilute solutions were obtained by using a Perkin Elmer Lambda 950 UV/VIS/NIR spectrophotometer. The molar absorption coefficients (ε) were calculated by applying the Lambert–Beer law to low absorbance spectra (A < 1) and recorded at successive dilutions.

Steady-state photoluminescence spectra were measured in right angle mode using an Edinburgh FLS920 fluorimeter, equipped with a Xenon arc lamp and a Hamamatsu R928P Peltier-cooled photomultiplier tube. The concentration of sample solutions was adjusted to obtain absorption values of A < 0.1 at the excitation wavelength. The solutions were de-aerated by bubbling argon for at least 20 min in custom-made gas-tight cuvettes. All emission spectra were corrected for the wavelength-dependent phototube response between 200 and 900 nm, using a calibration curve provided by the manufacturer. The luminescence quantum yields in the solution were evaluated by comparing the wavelength-integrated intensities of corrected spectra, with reference to [Ru(bpy)_3_]Cl_2_ (ϕ_r_ = 0.040 in air-equilibrated H_2_O) and quinine sulphate (ϕ_r_ = 0.53 in air-equilibrated H_2_SO_4_ 0.1 N) standards [54]. The phosphorescence spectra of the ligand in solvent-diluted glassy solutions at 77 K were recorded in gated detection mode on the same Edinburgh fluorimeter equipped with a pulsed Xe lamp.

The luminescence lifetimes were obtained using a TCSPC apparatus (HORIBA) equipped with a TBX Picosecond photon detection module and NanoLED/SpectraLED pulsed excitation sources. The analysis of luminescence decay profiles against time was accomplished using the Decay Analysis Software DAS v6.5 (HORIBA).

## Data Availability

Not applicable.

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
