# Peer review of "Rational Design of Mono- and Bi-Nuclear Cyclometalated Ir(III) Complexes Containing Di-Pyridylamine Motifs: Synthesis, Structure, and Luminescent Properties"

_molecules, 2022, doi:10.3390/molecules27186003_

Round 1

Reviewer 1 Report

This manuscript by Sesolis et al. reports the mono- and bi-nuclear Ir and Rh complexes and their luminescent properties. Luminescent properties of four complexes are well described. Investigation of the influence of different di-pyridylamine ligands to electronic and optical properties is of interest to the researchers working on the phosphorescent of metal complex. The referee believe that the article is worth being published in Molecules after minor revision.

The authors stated “while the Rh(III) derivative 4 is non-emissive.” What makes Rh(III) derivative 4 non-emissive? Luminescence property of compound 4 is very different from compounds 1-3. It would be useful if the authors briefly discuss the origin of the difference.

The authors should define (C^N) and (N^N).

Author Response

This reviewer considers the work is of interest to researchers working in the area of phosphorescent metal complex. The referee believes that the article is worth being published in Molecules after minor revision.

1-The authors stated “while the Rh(III) derivative 4 is non-emissive.” What makes Rh(III) derivative 4 non-emissive? Luminescence property of compound 4 is very different from compounds 1-3. It would be useful if the authors briefly discuss the origin of the difference.

Rh(III) derivative 4 is non-emissive. This behavior has already been reported for the mononuclear analogous complex [Rh(ppy)2(DPA)][PF6] with hexafluorophosphate. We added the related reference in the revised version "Photophysical and electrochemical properties of new ortho-metalated complexes of rhodium(iii) containing 2,2-dipyridylketone and 2,2-dipyridylamine. An experimental and theoretical study" Dalton Trans., 2007, 3440-3449. On the other hand it should be mentioned that cyclometalated iridium complexes Ir(C^N)2(L-L), due to their high-spin orbit coupling, tend to emit from 3LC-3MLCT excited states please check (Inorg. Chem. 2021, 60, 13222) while the related rhodium complexes display ligand centered emissions 3LC, please check (Inorg. Chem. 1987, 26, 1323, Topics in Current Chemistry in Photochemistry and photophysics of coordination compounds: rhodium, 2007, pp. 215). We feel that the emission of the rhodium complex 4 might deactivate due to close proximity to 3MC dark states. We added the new references in the revised version.

2- The authors should define (C^N) and (N^N).

Reviewer 2 Report

H. Amouri et al describe mono- and binuclear cyclometalated Ir(III) complexes and their luminescence properties. This paper should be revised the following issues for acceptance.

1) Complex 1, [Ir(ppy)2(DPA)][OTf], is almost same to that of [Ir(ppy)2(HDPA)](PF6) (ppy = 2-phenylpyridine, HDPA = 2,20-dipyridylamine), except for just anion (cited as ref 23). And the photoluminescence data of complex 1 in solution at room temperature and 77 K are very similar to that of [Ir(ppy)2(HDPA)](PF6). Thus, I am wondering if what is the significance of 1. That is, the data can be a kind of duplication. Please mention it.

2) Complexes 3 and 4 did not report the crystal structures. For exact characterization, it absolutely needs some ESI+ spectra. Please supply them.

3) In the sentence ”The phosphorescence emission of complex 2, bearing the phenyl-iodine group in the dipyridylamine ligand, is only slightly quenched by oxygen, suggesting that the excited state de-activation is dominated by the non-radiative intersystem crossing processes promoted by the strong spin-orbit coupling induced by the iodine atom.”, the spin-orbit coupling by I atom could be explained for better understanding.

4) In Table 2, the life times between RT and 77 K are quite different. Please mention about that in more detail for comparison.

5) The abbreviation “ppy” should be added their full name in Abstract.       

Author Response

This reviewer recommends publication after addressing the following issues mentioned below.

1-Complex 1, [Ir(ppy)2(DPA)][OTf], is almost same to that of [Ir(ppy)2(HDPA)](PF6) (ppy = 2-phenylpyridine, HDPA = 2,20-dipyridylamine), except for just anion (cited as ref 23). And the photoluminescence data of complex 1 in solution at room temperature and 77 K are very similar to that of [Ir(ppy)2(HDPA)](PF6). Thus, I am wondering if what is the significance of 1. That is, the data can be a kind of duplication. Please mention it.

Complex 1 contains a triflate anion and not a hexafluorophosphate as reported previously. As mentioned in the introduction we envisioned to probe the effect of dipyridylamine on the luminescent properties of mononuclear and dinuclear cyclometalated complex. To this end, complex 1 was synthesized to measure the quantum yield in order to compare it with the other parent mono- and di-nuclear complexes synthesized in this paper. Having 1 at hand, it was a useful benchmark for the assessment and interpretation of the photophysical behavior of complexes 2-4 in the same experimental condition. Moreover, the quantum yield of 1 has been mistakenly reported in the first version of the manuscript. It has been substituted for the correct value in the revised version.

2-Complexes 3 and 4 did not report the crystal structures. For exact characterization, it absolutely needs some ESI+ spectra. Please supply them.

Agreed. We have done the electrospray mass-spectrometry measurements for the rhodium and iridium dimer complexes 3 and 4. The results are positive and confirm the nature of the dimer species in solution. This important result was added in the revised version.

3-In the sentence “The phosphorescence emission of complex 2, bearing the phenyl-iodine group in the dipyridylamine ligand, is only slightly quenched by oxygen, suggesting that the excited state de-activation is dominated by the non-radiative intersystem crossing processes promoted by the strong spin-orbit coupling induced by the iodine atom.”, the spin-orbit coupling by I atom could be explained for better understanding.

The original sentence “The phosphorescence emission of complex 2, bearing the phenyl-iodine group in the dipyridylamine ligand, is only slightly quenched by oxygen, suggesting that the excited state de-activation is dominated by the non-radiative intersystem crossing processes promoted by the strong spin-orbit coupling induced by the iodine atom (z = 5,069 cm-1).” has been completed as “The phosphorescence emission of complex 2, bearing the phenyl-iodine group in the dipyridylamine ligand, is only slightly quenched by oxygen, suggesting that the excited triplet state de-activation back to the ground singlet state is dominated by the non-radiative intersystem crossing processes promoted by the strong spin-orbit coupling induced by the iodine atom (x = 5,069 cm-1) rather than the triplet-triplet energy transfer to the molecular oxygen.”

4-In Table 2, the life times between RT and 77 K are quite different. Please mention about that in more detail for comparison.

The following sentence, detailing the effect of temperature on the lifetime, has been added “This feature is commonly observed in Ir(III) ortho-metalated complexes, as the rigid matrix and the low temperature usually hinder the non-radiative mechanisms, thus increasing the luminescence intensity and the relevant excited state lifetime.

5-The abbreviation “ppy” should be added their full name in Abstract.

Agreed and done.    

Round 2

Reviewer 2 Report

The paper is well revised, so it could be accepted.